# Characterization of Vaginal Microbiota in Third Trimester Premature Rupture of Membranes Patients through 16S rDNA Sequencing

**DOI:** 10.3390/pathogens11080847

**Published:** 2022-07-28

**Authors:** Lou Liu, Jiale Chen, Yu Chen, Shiwen Jiang, Hanjie Xu, Huiying Zhan, Yongwei Ren, Dexiang Xu, Zhengfeng Xu, Daozhen Chen

**Affiliations:** 1Department of Obstetrics, The Affiliated Wuxi Maternity and Child Health Care Hospital of Nanjing Medical University, Wuxi 214002, China; wxliulou@njmu.edu.cn (L.L.); xuhanjie1995@126.com (H.X.); wxzhy2013@126.com (H.Z.); 2School of Public Health, Anhui Medical University, Hefei 230001, China; chenjiale98lele@163.com (J.C.); xudex@126.com (D.X.); 3Department of Research Institute for Reproductive Health and Genetic Diseases, The Affiliated Wuxi Maternity and Child Health Care Hospital of Nanjing Medical University, Wuxi 214002, China; 7211505001@stu.jiangnan.edu.cn (Y.C.); jiangsw137@126.com (S.J.); yongwei_ren@126.com (Y.R.); 4Department of Prenatal Diagnosis, Women’s Hospital of Nanjing Medical University, Nanjing Maternity and Child Health Care Hospital, Nanjing 210004, China

**Keywords:** premature rupture of membranes, 16S rDNA sequencing, vaginal dysbiosis, *Lactobacillus*

## Abstract

In China, premature rupture of membranes (PROM) counts as a major pregnancy complication in China and usually results into adverse pregnancy outcomes. We analysed the vagina microbiome composition using 16S rDNA V3–V4 amplicon sequencing technology, in this prospective study of 441 women in their third trimester of pregnancy. We first divided all subjects into PROM and HC (healthy control) groups, in order to investigate the correlation of vagina microbiome composition and the development of PROM. We found that seven pathogens were higher in the PROM group as compared to the HC group with statistical significance. We also split all subjects into three groups based on *Lactobacillus* abundance-dominant (*Lactobacillus* > 90%), intermediate (*Lactobacillus* 30–90%) and depleted (*Lactobacillus* < 30%) groups, and explored nine pathogenic genera that were higher in the depleted group than the intermediate and dominant groups having statistical significance. Finally, using integrated analysis and logistics regression modelling, we discovered that *Lactobacillus* (coeff = −0.09, *p* = 0.04) was linked to the decreased risk of PROM, while *Gardnerella* (coeff = 0.04, *p* = 0.02), *Prevotella* (coeff = 0.11, *p* = 0.02), *Megasphaera* (coeff = 0.04, *p* = 0.01), *Ureaplasma* (coeff = 0.004, *p* = 0.01) and *Dialister* (coeff = 0.001, *p* = 0.04) were associated with the increased risk of PROM. Further study on how these pathogens interact with vaginal microbiota and the host would result in a better understanding of PROM development.

## 1. Introduction

Premature rupture of membranes (PROM) is characterized as spontaneous membranes’ rupture that occurs before the onset of labour [1]. The occurrence of membranes’ rupture prior to 37 gestational weeks has been referred to as preterm PROM (pPROM). Ascending vaginal infection is the main cause of PROM. One significant characteristic of pregnancy vaginal infection is a mixed infection, which often contributes to approximately 30% reproductive tract infection cases [2]. But a lot of vaginal infections and cases of vaginal microbiome dysbiosis are asymptomatic. George et al. [3] have shown that ascending infection from vaginal to amnionic membranes could induce PROM even in asymptomatic vaginal dysbiosis. Only 1% of the pathogens in the vaginal microenvironment may be detected by microscopic inspection or a bacteria culture test [4]. Thus, the mixed and asymptomatic vaginal infection usually cannot be treated in a timely manner. Aggressive vaginal infection by *Peptostreptococcus*, *Streptococcus* and *Ureaplasma* among others might release endotoxin (typically LPS) [5] and induce vaginal epithelia to secrete inflammatory cytokines such as IL-6, and activate matrix metalloproteinases-8 (MMP-8) on the fetal membranes, which results in the breakdown of the extracellular matrix, weakens the fetal membrane, and eventually results in membrane rupture [6]. Negative pregnancy outcomes are typically caused by microbiome dysbiosis linked to mixed vaginal infections including group B streptococcus infection [6] and vulvovaginal candidiasis [7]. De Seta F et al. reported that in pPROM, the primary vaginal pathogens were *Candida albicans*, *Trichomonas vaginalis*, and *Gardnerella* species [7]. Vaginal lactobacilli maintain the balance of the local microbial environment by reducing vaginal pH through the production of lactic acid and decrease the risk of pregnancy complications [8].

In the United States, the preterm birth incidence is about 10%, and pPROM occurs in approximately 30% of preterm birth cases [9]. Preterm birth alone is predicted to cost more than $26.2 billion annually [10]. The incidence of the preterm term is highly linked to race; for instance, the rate among Non-Hispanic Black women was 13.4% compared to 9.6% in all women according to Wheeler et al. report [11]. Due to China’s slightly different definition of preterm birth (delivery between 24 and 36^+6^ gestational weeks), plus the popularity of tocolytics in clinical practice, the preterm birth rate had been reported as low as 6.9% in a recent epidemiology study in 2015–2016 [12]. Xia et al. [13] carried out another large population study and discovered 17049 (15.3%) women with PROM and among which 13,927 (12.5%; and 81.7% of all PROM cases) were term PROM. According to these studies, although the PROM rate is high, the preterm PROM rate is comparatively low in the Chinese population (2.6%) [14] in contrast to what has been reported in United States (3–3.25%) [15], yet, the unfavourable pregnancy outcomes are comparable. On the other hand, years of excessive antibiotics usage in China resulted in the establishment of certain resistant bacterial strains, such as *Staphylococcus aureus* [16]. Although obstetricians often strictly follow the guidelines explained in “Practice Bulletin: Premature Rupture of Membranes”, reproductive tract infection such as chorioamnionitis, fetal infection, and neonatal sepsis could be caused by those resistant bacterial strains. The combination of all these variables makes foetal membrane rupture during the late stages of pregnancy a frequent occurrence. Apparently, attention needs to be paid to prevent severe adverse complications from these pregnant women and neonates.

Recently, culture-independent approaches, which includes 16S amplicon sequencing, were applied in order to examine the pathogens in mixed vaginal infection. Brown et al. [17] studied vaginal microorganism composition at four pregnancy stages (12–17^+6^, 18–23^+6^, 24–29^+6^ and 30–36^+6^ gestational weeks) and explored that vaginal dysbiosis at 24–29^+6^ and 30–36^+6^ weeks significantly enhanced the risk of PROM. Therefore, they called these two stages as “immune clock”. This paper states that the purpose of this prospective study was to characterize the differences in vaginal microbiota composition between the PROM group and the healthy control (HC) group. We aimed to discover the primary pathogens and the features of vaginal microbiome composition in PROM group through 16S rDNA amplicon sequencing of vaginal swabs in the third trimester during 31–36 gestational weeks, and explore their correlation to the development of PROM.

## 2. Results

### 2.1. Demographical and Clinical Data

This study originally included 441 individuals in total, of which 15 cases were withdrawn owing to loss to follow-up. Among the final 426 subjects, 84 women developed PROM and the incidence of PROM was 19.7%. To match the mother age and gestational weeks at sampling and to minimize the impact of ageing and hormones, we chose 45 PROM patients and 90 HC cases for the 16S rDNA sequencing analysis. Maternal age, nulliparity, white blood cell at admission, positive genital cultures at admission, gestational age at sampling, steroid administration, tocolysis treatment, gestational age at delivery, latency from sampling to delivery, BMI in the first trimester, BMI in third trimester, maternal serum estriol at sampling, baby weight at birth, baby gender, number of deliveries, and spontaneous abortion were among the clinical characteristics gathered. The clinical characteristics from PROM and HC groups matched well, and there were no gestational diabetes or preeclampsia cases in the study cohort (Table 1).

### 2.2. 16S rDNA V3–V4 Gene Sequencing Statistics

The mean reads for each sample was 63,085.1 (ranging from 39,104 to 68,989, Appendix A; and most samples had >50,000 reads, Appendix A); and a total of 9,462,772 high-quality reads had passed quality control. The species dilution curve demonstrated that the total samples were enough for the comparison (Appendix A). Additionally, the Sobs diversity rarefaction curve made sure that each sample’s sequencing depth was enough for the study’s goals (Appendix A).

### 2.3. Different Microbiome Composition between PROM and HC Groups

All vaginal swap samples yielded a total of 1445 OTUs of which 351 were shared by the PROM and HC groups, while 330 and 764 unique OTUs were explored in the PROM and HC groups, respectively (Appendix A). Regarding the taxonomic compositions of the microbiomes, the Principal Coordinates Analysis (PCoA) indicated no statistically significant difference between the two groups (Appendix A). In comparison to the HC group, the PROM group’s Shannon and Simpson indices were greater (both *p* = 0.014; see Appendix A). These findings indicated that the alpha-diversity of microbiome taxonomy in PROM group was higher than in HC group. Nevertheless, the Chao index displayed no statistical difference between the two groups (Appendix A). The microbiome community richness did not significantly differ between the two groups, according to the dimensionality reduction and beta-diversity analyses. We had difficulty in determining approximately 50% of the species when using QIIME 2.0 software in order to assess microbiome composition at the ASV-level, therefore we only compared microbiome composition differences at higher levels. As can be shown in Figure 1a, samples from the PROM group showed greater genus-level total and relative abundances of vaginal pathogens than samples from the HC group. Comparatively, the relative abundance of *Lactobacillus* in the HC group had been higher than that in the PROM group (*p* = 0.036; Figure 1b). Among the top 15 most abundantly detected pathogens, 7 pathogens in total were statistically significantly higher in the PROM group than in the HC group. These pathogens included *Gardnerella* (*p* = 0.02), *Megasphaera* (*p* = 0.01), *Prevotella* (*p* = 0.02), *Ureaplasma* (*p* = 0.01), *Dialister* (*p* = 0.04), *Aerococcus* (*p* = 0.01) and *Arcanobacterium* (*p* = 0.03) (Figure 1c). At the phylum level, we also observed significantly higher pathogen abundance in the PROM group, which includes *Veillonellales Selenomonadales* (*p* = 0.01), *Mycoplasmatales* (*p* = 0.01), *Bacteroidales* (*p* = 0.02), and *Bifidobacteriales* (*p* = 0.02), while the relative abundance of *Lactobacillales* had been lower in the PROM group than in the HC group (*p* = 0.04; Figure 1d). Together, these findings proposed that, in the PROM group, the relative abundance of *Lactobacillales* decreased, while the total number and relative abundance of other pathogens increased. To find possible biomarkers of PROM incidence, we also used linear discriminant analysis (LDA). The LDA score (log 10) also showed that *Lachnospiraceae*, *Eggerthellaceae*, *Aerococcaceae*, *Aerococcus*, *Romboutsia* and *Klebsiella* were enriched in the PROM group while *Lactobacillus* was primarily enriched in the HC group (Figure 1e).

### 2.4. Bioinformatics Analysis among Groups with Different Abundance of Lactobacillus

According to the above-mentioned findings, the abundance of *Lactobacillus* decreased, while the diversity as well as relative abundance of pathogenic bacterial taxa enhanced the PROM group. *Lactobacillus* is known to play a vital protective role in the healthy vaginal microenvironment. Thus, we also utilized *Lactobacillus* abundance to re-divide all 135 subjects into three groups: (1) the dominant group (*Lactobacillus* > 90%), (2) the intermediate group (*Lactobacillus* 30–90%) the depleted group (*Lactobacillus* < 30%) [18], and investigated how different infections affected the incidence of PROM in these populations. Among the top 15 relatively abundant pathogens at the genus level, a total of nine pathogenic genera had been higher in the depleted group as compared to the intermediate and dominant groups with statistical significance, which includes *Fannyhessea* (previously known as *Atopobium*, *p* < 0.0001), *Megasphaera* (*p* < 0.0001), *Gardnerella* (*p* < 0.0001), *Prevotella* (*p* < 0.0001), *Shuttleworthia* (*p* < 0.0001), *Sneathia* (*p* < 0.0001), *Dialister* (*p <* 0.0001), *Aerococcus* (*p* < 0.0001), and *Veillonella* (*p* < 0.0001) (Figure 2a).

Then, we utilized a cladogram graph in order to analyze the microbiome taxa that might play an essential role in these three groups (Figure 2b). In the dominant group, the p functional microbiome was *Lactobacillus*, compared to *Gardnerella*, *Fannyhessea*, *Coriobacteriaceae*, *Prevotella*, *Aerococcus*, *Dialister*, *Megasphaera* and *Sneathia* in the intermediate group, and *Corynebacterium*, *Streptococcus*, *Anaerococcus*, *Finegoldia* and *Veillonella* in the depleted group. In general, the dominant group’s microbiome taxa were lower than those in the intermediate group and the deficient group. The LDA score (log 10) also proposed that *Lactobacillus* was enriched in the dominant group, *Streptococcus* had been enriched in the intermediate group, while *Gardnerella*, *Fannyhessea*, *Megasphaera*, *Prevotella* were enriched in the depleted group (Figure 2c).

### 2.5. Cross-Analysis between the Two Grouping Methods

In this study, we examined the role of the pathogenic microbiome in the development of PROM by categorizing all patients into groups based on several characteristics. The first grouping method had been based on pregnancy outcomes: the PROM group and HC group. The second grouping method was based on the abundance of *Lactobacillus*: the dominant group, the intermediate group, as well as the depleted group. Based on the above-mentioned findings, the microbiome composition had been similar in all groups, although their relative abundances varied a lot. As a result, we further analysed the relationship between relative microbiome abundance as well as the occurrence of PROM in three groups.

The characteristics and diversity of vaginal microbiome compositions among the dominant group, intermediate group, and depleted group had been demonstrated as following. The Venn diagram representing the three groupings of OTUS was used (Appendix A). The Principal Coordinates Analysis (PCoA) (Appendix A) as well as the Shannon and Simpson index (Appendix A) showed a statistically significant difference (*p* < 0.001) among the three groups. There had been a statistically significant difference between the intermediate and dominant groups for The Chao index (*p* = 0.02; see Appendix A). Results of Principal Component Analysis (PCA) and Non-metric Multidimensional Scaling (NMDS) with the Adonis test were displayed in Appendix A. The ace index as well as species richness index displayed a statistically significant difference between the intermediate and dominant groups (*p* = 0.07 and *p* = 0.04, respectively; see Appendix A). These findings showed that the intermediate group had a higher richness and greater diversity of bacteria than the dominating group. In the meantime, the alpha-diversity and beta-diversity of microbiome taxonomy were higher in the depleted group as compared to the intermediate and dominant groups.

Interestingly, the dominating group had much fewer PROM cases (31 PROM *versus* 90 HC cases, *p* < 0.001) than the depleted group (4 PROM versus 3 HC cases in the depleted group, and 10 PROM versus 7 HC cases in the intermediate group) (Figure 3a). This observation further supported the protective role of *Lactobacillus* in the development of PROM. When comparing microbiome composition, both pathogen abundance and diversity were higher in the depleted group and the intermediate group as compared to the dominant group. Moreover, they were also higher in the PROM group as compared to the HC group (Figure 3b). In order to further identify these pathogens, their relative abundance had been compared between PROM cases and HC cases in the dominant group, the intermediate group as well as the depleted group, respectively. In the dominant group, there seemed to be difference of the abundance of *Prevotella* and *Ureaplasma* between PROM and HC cases, the relative low quantity of the pathogens in these situations, however, is likely to be the reason why these changes did not achieve statistical significance (Figure 3c). Nevertheless, in the intermediate group, the relative abundance of *Ureaplasma* and *Megasphaera* was higher in PROM cases than in HC cases (*p* = 0.032 and *p* = 0.039, respectively; Figure 3d). Similar to the depleted group, the *Prevotella* had a statistically higher relative abundance in PROM cases than in HC cases (*p* = 0.015; Figure 3e). The combined findings provided compelling evidence that the loss of *Lactobacillus* was related with bacteria richness and diversity in vaginal microbiome, as well as the occurrence of PROM. Furthermore, the high abundance of *Ureaplasma*, *Megasphaera* and *Prevotella* in *Lactobacillus* low cases might have a correlation to the development of PROM which warrants further investigation in animal models along with the follow up clinical studies.

### 2.6. Intergrated Analysis of Vaginal Microbiome in All Samples

The cluster heat map for the three groups revealed that the depleted and intermediate groups had higher expression levels of vaginal bacteria including *Megasphaera*, *Fannyhessea*, *Shuttleworthia*, *and Gardnerella*. Comparatively, *Lactobacillus* was highly expressed in the dominant group (Figure 4). In the logistic regression model analysis, 6 microbial genera had correlation the with PROM occurrence, among which *Lactobacillus* (coeff = −0.09, *p* = 0.04) was primarily linked to the decreased risk of PROM, while *Gardnerella* (coeff = 0.04, *p* = 0.02), *Prevotella* (coeff = 0.11, *p* = 0.02), *Megasphaera* (coeff = 0.04, *p* = 0.01), *Ureaplasma* (coeff = 0.004, *p* = 0.01) and *Dialister* (coeff = 0.001, *p* = 0.04) were associated with the increased risk of PROM (Table 2).

## 3. Discussion

The vaginal microbiota is crucial for a healthy woman’s reproductive system. Typically, *Lactobacillus* species, such as *L. iners*, *L. crispatus*, *L. gasseri* and *L. jensenii* dominates the vaginal bacterial populations [19]. PROM is often linked to the ascending vaginal infection which disrupts the healthy vaginal microbiome. In this study, we applied 16S rDNA V3–V4 amplicon sequencing technology in order to compare the microbiome composition between the PROM group and HC group. According to reports, the bacterial 16S region’s V1–V3 area can be used to detect the makeup of the *Lactobacillus* population [20], meanwhile, the V4 variable region of the 16S gene provides intense discrimination among most bacteria species [21]. Thus, the V3–V4 region of the 16s gene we selected in this study must enhance the microbiome detection coverage. Bieger [22] et al. contrasted the results from 16S rRNA amplicon sequencing, as well as whole-genome shotgun sequencing, and found that 16S region amplicon sequencing was unable to identify the whole microbiome at the species level. Therefore, it is explained in the high percentage of the unclassified microbiome at the species level in our study. This restriction forced us to focus our analysis on microbiome composition at or above the genus level. According to our results, the relative abundance of *Lactobacillus* was much lower in the PROM group than in the HC group (*p* = 0.04; Figure 1b), which proposed that lacking of protection from *Lactobacillus* was one of the major causes of PROM occurrence. Meanwhile, the relative abundance of other pathogenic bacteria increased in the PROM group compared to the HC group, which includes *Gardnerella* (*p* = 0.02), *Megasphaera* (*p* = 0.01), *Prevotella* (*p* = 0.02), *Ureaplasma* (*p* = 0.01), *Dialister* (*p* = 0.04), *Aerococcus* (*p* = 0.01), and *Arcanobacterium* (*p* = 0.03) at the genus level, and all the differences reached statistical significance (Figure 1c). We also noticed a considerably greater pathogen abundance in the PROM group at the phylum level, including *Veillonellales Selenomonadales* (*p* = 0.01), *Mycoplasmatales* (*p* = 0.01), *Bacteroidales* (*p* = 0.02), and *Bifidobacteriales* (*p* = 0.02; Figure 1d). *Gardnerella* is a predominant vaginal bacteria species and was reported to be related to bacterial vaginosis and adverse pregnancy outcomes, which includes pPROM [23]. Tabatabaei et al. studied vaginal microbiome in early pregnancy and found *Gardnerella* had a positive relationship with spontaneous preterm birth [24]. Moreover, Jayaprakash et al. reported that *Prevotella* spp. and *Megasphaera* type I were ubiquitously found in pPROM patients [25]. It is noteworthy that the pathogens found in pPROM, spontaneous preterm birth, and PROM subjects had been very similar, thus, all these conditions are likely caused by same mechanism-reproductive tract infection.

We re-divided all cases into three groups based on the presence of *Lactobacillus* in order to further investigate the relationship between the lower relative abundance of *Lactobacillus* spp. and the emergence of PROM. Interestingly, when the relative abundance of *Lactobacillus* increased, the incidence of PROM significantly fell. PROM incidence was much lower in the dominant group (31 out of 121 cases) than in the intermediate and depleted groups (Figure 3a). It had been noteworthy that 16S rDNA sequencing technology is not capable of clearly differentiating different *Lactobacillus* species, while those different *Lactobacillus* species might play very different role in vaginal microbiota as well as the consortia formed by different lactobacilli can also influence their function. Pacha-Herrera et al. [24] identified *Lactobacillus* species by PCR and found that *L. acidophilus* demonstrated statistically significant differences in its prevalence on healthy microbiota against both dysbioses (bacterial vaginosis, *p* = 0.041; and aerobic vaginitis, *p* = 0.045), whereas *L. jensenii* showed statistically significant differences between healthy microbiota and aerobic vaginitis cases (*p* = 0.012). This is most likely one of the factors contributing to the approximately one-fourth of participants in the dominant group who still had PROM. The vaginal microbiome of reproductive-age women based on *Lactobacillus* could be split into five types: type I, type II, type III, type V are dominated by *L. crispatus*, *L. gasseri*, *L. iners*, and *L. jensenii*, respectively, while type IV does not have dominant *Lactobacillus* but higher proportions of strictly anaerobic bacteria [19]. *Lactobacillus* could compete with pathogenic bacteria in order to adhere to host cells, form a physical barrier, and bind to Toll-like receptors on the host cell surface for further activating signal pathways in host cells and increase the secretion of anti-inflammatory factors through the S-layer protein [26]. *L. crispatus* possesses the strongest protective vagina milieu because of its S-layer protein [27]. Contrarily, vaginal dysbiosis frequently results in a rise in the high abundance of L. iners [28]. Detailed characterization of *Lactobacillus* species in these dominant group subjects using other methods, such as whole genome sequencing, might refine the biomarkers identification for PROM further.

We also discovered certain pathogens were correlated to the occurrence of PROM, when comparing the PROM and HC cases among the dominating, intermediate, and depleted groups, respectively. For example, in the intermediate group, the relative abundance of *Ureaplasma* and *Megasphaera* was higher in PROM cases than in HC cases (*p* = 0.032 and *p* = 0.039, respectively; Figure 3d). In the depleted group, the *Prevotella* showed higher relative abundance in PROM cases than in HC cases (*p* = 0.015; Figure 3e). In our previous grouping study, these pathogens were also found higher in the PROM group as compared to the ones in the HC group, as per our earlier grouping analysis (Figure 1c). All these results together showed that the high abundance of *Ureaplasma*, *Megasphaera*, and *Prevotella* may correlate to the development of PROM. Lee et al. assessed vaginal swabs gained from 1035 pregnant women and detected *Ureaplasma* in 472 specimens, as well as preterm premature rupture of the membrane cases were often found in the *Ureaplasma* infected patient [29]. It is interesting to note that the genome of the *Ureaplasma* spp. has numerous genes that code for surface proteins, with the Multiple Banded Antigen gene serving as the most crucial one (MBA). The C-terminal domain of MBA is antigenic and elicits a host antibody response and can cause genital tract infections, and severe genital tract infections can cause the PROM [30].

It’s interesting that some individuals with normal vaginal microbiome composition could still develop PROM (Figure 3a, dominant group). We hypothesized that this may be because each person’s local host mucosal immune response may be very different, which could either induce or inhibit PROM incidence [31,32]. We further performed heatmap analysis as well as logistic regression modelling for assessing all the 135 samples together and assessed which microbiota had a relatively strong influence on the PROM. *Gardnerella* (coeff = 0.04, *p* = 0.02), *Prevotella* (coeff = 0.11, *p* = 0.02), *Megasphaera* (coeff = 0.04, *p* = 0.01), *Ureaplasma*(coeff = 0.004, *p* = 0.01) and *Dialister* (coeff = 0.001, *p* = 0.04) were associated with the increased risk of PROM (Figure 4), which is consistent with what has been reported before [33,34].

Collectively, the prevalence of PROM is linked to decreased *Lactobacillus* spp. abundance and increased variety and quantity of pathogenic bacteria. *Lactobacillus* has a protective role to play in maintaining a healthy local vaginal milieu by inhibiting the growth of other bacteria. Vaginal dysbiosis might inflict an interaction between the local mucosal immunity and bacteria, which results into the development of PROM if the balance of vaginal microbiota is severely destroyed and cannot be reversed by the host immune system. The primary pathogens linked to PROM are *Gardnerella*, *Prevotella*, *Megasphaera*, and *Ureaplasma*. Further study on how these pathogens interact with vaginal microbiota and the host will shed light on a clearer understanding of PROM development.

Limitations of the study. We used 16S rDNA amplicon sequencing technology in order to assess the microbiome composition difference between the PROM and HC groups, and identified potential biomarkers for evaluating the risk of developing PROM. The research does have a few drawbacks, though. First, it is impossible to tell from the sequencing approach if the detected microbes are alive and physiologically active. Second, despite the fact that some of the microorganisms could be crucial role as biomarker of PROM, but had not been called out by the bioinformatics analysis because of the lack of statistical significance. Third, the chance of development into PROM, due to the cross-talk between bacteria and vaginal immunity, might be different in different patients even though they share similar microbiome composition. All these limitations warrant further validation of our findings in relevant in vivo models.

## 4. Materials and Methods

### 4.1. Subjects

Regardless of their clinical risk factors for PROM, a total of 441 pregnant women were enlisted from the Obstetric Department of the Affiliated Wuxi Maternity and Child Health Care Hospital of Nanjing Medical University (from 18 September 2019 to 21 November 2020). Table 1 lists the age range of these pregnant ladies. The Affiliated Wuxi Maternity and Child Health Care Hospital of Nanjing Medical University’s ethics committee granted approval for the study (No. 2019-02-0402-01) and the present study was conducted following the Declaration of Helsinki Principles. Retrospectively, this study has been registered in the Chinese Clinical Trial Registry system (ChiCTR2000034721, registration date: 16 July 2020, https://www.chictr.org.cn/showproj.aspx?proj=56635, accessed on 18 July 2020). All the participants in this study provided their written, informed consent. The prevalence of PROM in current study is 19.05% and we utilized PASS 11.0 software (One-Sample Sensitivity and Specificity Power Analysis) to estimate the sample size required to draw clear conclusion (α = 0.05, 1 − β = 0.8, sensitivity = 72%, specificity = 75%). A two-sided binomial test with a total sample size of 220 participants, including 42 subjects with PROM, has 100% power to detect a change in specificity power to identify a change in sensitivity from 0.5 to 0.72. In the current study, the sequencing analysis comprised 90 matched healthy control cases and 45 PROM patients. The actual significance level acquired by the sensitivity test is 0.0436, and achieved by the specificity test is 0.0427 (the target significance level is 0.05).

### 4.2. Prenatal Examination and Delivery

According to WHO’s antenatal care guidelines, all enrolled pregnant women underwent routine prenatal examinations (in the first 12 gestational weeks, with subsequent follow-ups at 20, 26, 30, 34, 36, 38, and 40 gestational weeks, at least eight times during the gestational period). Between 31–36 weeks of pregnancy, sample swabs were taken from posterior fronix and promptly snap-frozen in liquid nitrogen before being preserved at −80 °C. All pregnant women delivered babies in the Affiliated Wuxi Maternity and Child Health Care Hospital of Nanjing Medical University, as well as the pregnant women and neonatal outcomes were recorded for statistics.

### 4.3. PROM Cases and Healthy Controls Selection

According to the guidelines from “ACOG Practice Bulletin No. 188: Prelabor Rupture of Membranes”, PROM was diagnosed [22]. The pregnant women who met the criteria of PROM had been assigned to the PROM group. The healthy control (HC) group was made up of additional pregnant women who went into delivery without PROM. Two HC control samples and one PROM sample were matched for age as well as sampling time at gestational weeks in order to reduce hormone influence at different ages and gestational weeks. 90 HC cases and 45 PROM cases altogether were selected for further study. PROM cases with gestational diabetes mellitus or preeclampsia were excluded from the study to reduce the influence from causes other than vaginal microbiome dysbiosis. Details can be found in the flowchart for patient selection (Appendix A).

### 4.4. DNA Extraction

In accordance with the Ravel et al. technique, DNA was extracted from vaginal swaps [19]. Briefly, using a PowerSoil DNA Isolation kit from (MOBIO, Carlsbad, CA, USA) and following the manufacturer’s instructions, DNA was extracted. Lysozyme had been used to fully lyse Gram-positive bacteria and ensure the integrity of library. We utilized six swabs as sample blanks. Total DNA from the swab was eluted in 20 µL of elution buffer (Omega D3096, Norcross, GA, USA) and stored at −80 °C until amplification by polymerase chain reaction (PCR). Utilizing electrophoresis on a 1 percent agarose gel, the purity of the genomic DNA was verified. In order to avoid microorganism contamination [35], 6 sterilized blank vaginal swaps were exposed to air simultaneously as PROM group and HC group as negative control samples. From those blank exchanges, no microbial DNA was discovered. This phenomenon proved that environmental factors did not have any influence on the experimental results.

### 4.5. 16S rDNA Gene Sequencing and Data Processing

The V3–V4 region of the 16S rDNA genes had been amplified by PCR with a universal forward primer 16V3 (341F: 5′-ACTCCTACGGGAGGCAGCAG-3′) and a unique barcoded fusion reverse primer 16V4 (806R: 5′-GGACTACHVGGGTWTCTAAT-3′, where R indicates purine), which are specific for V3–V4 hypervariable regions of 16S rDNA [36]. Illumina adapter, pad and linker sequences were added to the forward and reverse primers, respectively. PCR amplification had been performed in a 50 μL reaction containing 30 ng template, fusion primer and PCR master mix (MicroSEQ™500 16S rDNA PCR kit, ThermoFisher Scientific, Waltham, MA, USA). Using the following conditions, PCR was performed: 3 min denaturation at 94 °C; 25 cycles of denaturation at 94 °C for 45 s, annealing at 50 °C for 60 s, elongation at 72 °C for 90 s; and a final extension at 72 °C for 10 min. For each sample, three replicates were carried out. The PCR products of the same sample had been mixed and confirmed by 2% agarose gel electrophoresis and purified with AmpureXP beads and eluted in Elution buffer, quantified by QuantiFluor™-ST Blue Fluorescence Quantitative System (Promega, Madison, WI, USA), and finally blended in accordance with each sample’s required sequencing volume. Using the Agilent 2100 bioanalyzer (Agilent, Santa Clara, CA, USA), libraries were measured. The validated libraries had been utilized for sequencing on Illumina MiSeq platform (BGI, Shenzhen, China) following the standard pipelines of Illumina, which generated 2 × 300 bp paired-end reads. Based on the overlapping regions, the paired-end sequencing data acquired from Miseq sequencing (Illumina, San Diego, CA, USA) were further merged into single sequences. The samples were demultiplexed and the sequence direction was determined using the barcodes and primer sequences. Reads quality control was set as following: mismatch rate in overlapping region < 0.2; max mismatch base-pairs in barcode region = 0; max mismatch base-pairs in primer region = 2. The software utilized for sequence processing and OTU clustering includes: Trimmomatic [37], FLASH [38], QIIME version 2.0 [39], Dereplication [40], and Singletons [41]. OTU clustering was performed with at least 97% similarity.

### 4.6. Taxonomic Assignment

Using RDP classifier Bayesian algorithm at a similarity level of 97% for all OTU representative sequences. Composition of each sample had been analysed at each classification level: phylum, class, order, family, genus and species. Silva138 was the comparison database used (database release 111 (July 2012)), https://www.arb-silva.de/documentation/release-1381/, 21 Febuary 2021) at the ASV level.

### 4.7. Statistical Analysis

Maternal characteristics had been analysed in the R environment. A chi-square test was used to compare categorical variables. Numeric variables were presented as average ± SEM, and compared by the independent-sample Mann-Whitney U test.

The Wilcoxon rank-sum test was performed to determine relative abundance in order to examine preliminary findings of the overall microbial structure and test the most frequent genera and alpha diversity. Clustering of microbial communities and between samples diversity (beta diversity) had been analysed using Bray–Curtis or Unifrac, and visualized by Principal Coordinates Analysis (PCoA) [42]. To identify PROM microbiological biomarkers, the Wilcoxon test was conducted to the differential relative abundance analysis utilising the raw (non-rarefied) dataset. The LEfSe method [43] had been used to assess differentially abundant taxonomic features between all groups. An alpha value of 0.05 was adopted for the factorial Kruskal–Wallis test between classes. A minimum threshold of 2.0 was utilized for logarithmic LDA (Linear Discriminant analysis) scoring of discriminative features.

## 5. Conclusions

Low levels of *Lactobacillus* and high levels and diversity of conditional pathogenic bacteria were connected with the characteristics of the vaginal microbiota in PROM cases. *Lactobacillus* protected a healthy local vaginal milieu by inhibiting the growth of bacteria. An interaction between the local mucosal immunity and bacteria may result from vaginal dysbiosis. If the balance of vagina microbiota was destroyed to a serious situation and cannot be reversed by host immune system, a PROM case is easily established among women. The main pathogens in the PROM group were Gardnerella, Prevotella, and Megasphaera at the genus level in this study, being statistically associated with PROM establishment among women. Further research, such as co-culture experiment using fetal membrane tissue and bacteria identified in this study, is needed to better understand how these conditional pathogens interact with the host and the vaginal microbiome and result in PROM.

## Figures and Tables

**Figure 1 pathogens-11-00847-f001:**
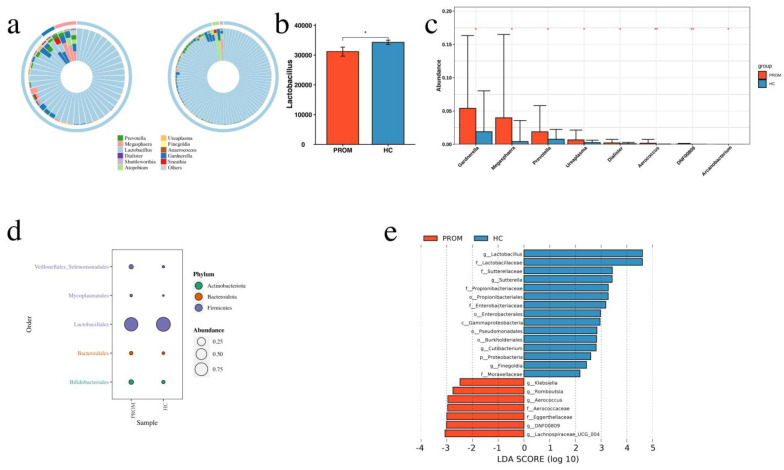
Comparison of vaginal microbiome composition between the PROM and HC groups. (**a**) The relative abundance of *Lactobacillus* had been lower, while that of other pathogens was higher in the PROM group than in the HC group. (**b**) Comparison of *Lactobacillus* relative abundance between the PROM and HC groups (*p* = 0.036). (**c**) Comparison of relative abundance of seven pathogens between the PROM and HC groups, including *Gardnerella* (*p* = 0.02), *Megasphaera* (*p* = 0.01), *Prevotella* (*p* = 0.02), *Ureaplasma (p* = 0.01), *Dialister* (*p* = 0.04), *Aerococcus* (*p* = 0.01), *Arcanobacterium* (*p* = 0.03). They were all statistically significant. (**d**) At the phylum level, relative abundance of four pathogens was higher in the PROM group than HC group with statistical difference, including *Veillonellales_Selenomonadales* (*p* = 0.01), *Mycoplasmatales* (*p* = 0.01), *Bacteroidales* (*p* = 0.02), *Bifidobacteriales* (*p* = 0.02). In comparison to the HC group, the relative abundance of Lactobacillales was lower in the PROM group (*p* = 0.04). (**e**) The LDA score (log 10) led by the PROM group showed that *Lachnospiraceae*, *Eggerthellaceae*, *Aerococcaceae*, *Aerococcus*, *Romboutsia*, *Klebsiella* were enriched in the PROM group, while *Lactobacillus* was enriched in the HC group. * *p* < 0.05, ** *p* < 0.01.

**Figure 2 pathogens-11-00847-f002:**
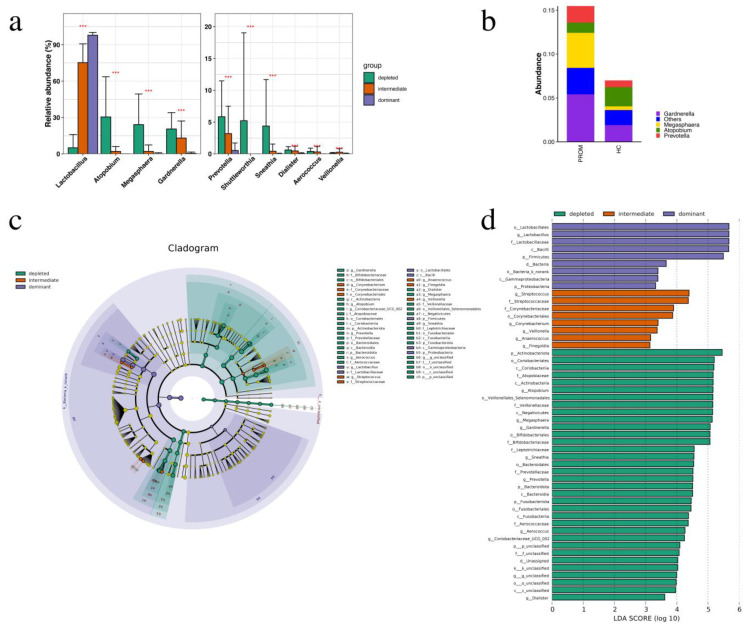
Comparison of the dominating group, the intermediate group, as well as the depleted group. (**a**) Pathogens showed higher relative abundance in the depleted group than the intermediate and dominant groups with a statistical significance included *Fannyhessea* (*p* < 0.0001), *Megasphaera* (*p* < 0.0001), *Gardnerella* (*p* < 0.0001), *Prevotella (p* < 0.0001), *Shuttleworthia* (*p* < 0.0001), *Sneathia* (*p* < 0.0001), *Dialister* (*p* < 0.0001), *Aerococcus* (*p* < 0.0001), *Veillonella* (*p* < 0.0001). (**b**) Pathogens showed higher relative abundance in the PROM group than the HC group. (**c**) Microbiome taxa had a significant impact on the three groups, according to the LEfSe study (**d**) The LDA score (log 10) displayed that *Lactobacillus* was enriched in the dominant group, *Streptococcus* was enriched in the intermediate group, while *Gardnerella*, *Fannyhessea*, *Megasphaera*, *Prevotella* were enriched in the depleted group, respectively. *** *p* < 0.0001.

**Figure 3 pathogens-11-00847-f003:**
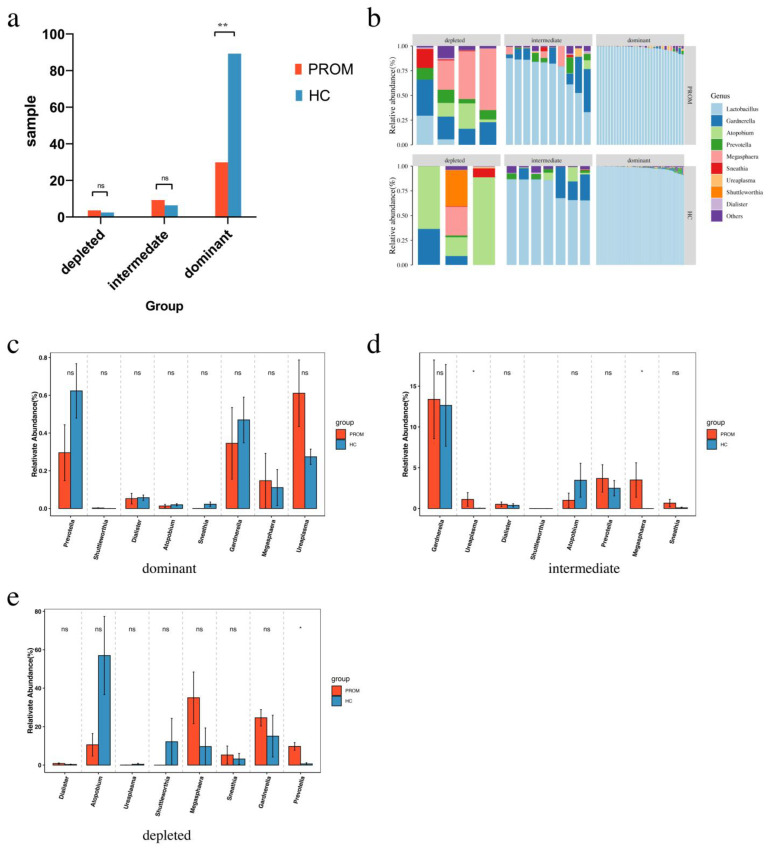
Cross-analysis between the two grouping methods. (**a**) HC and PROM case distribution throughout the three groups. (**b**) The relative abundance of pathogens in all groups. (**c**) The relative abundance of Ureaplasma had been higher in PROM cases than in HC cases in the dominant group, but the difference failed to reach statistical significance. (**d**) The relative abundance of Ureaplasma and Megasphaera was higher in PROM cases than in HC cases (*p* = 0.032 and *p* = 0.039, respectively) in the intermediate group. (**e**) The relative abundance of Prevotella was higher in PROM cases than in HC cases (*p* = 0.015) in the depleted group. ns *p* > 0.05, * *p* < 0.05, ** *p* < 0.01.

**Figure 4 pathogens-11-00847-f004:**
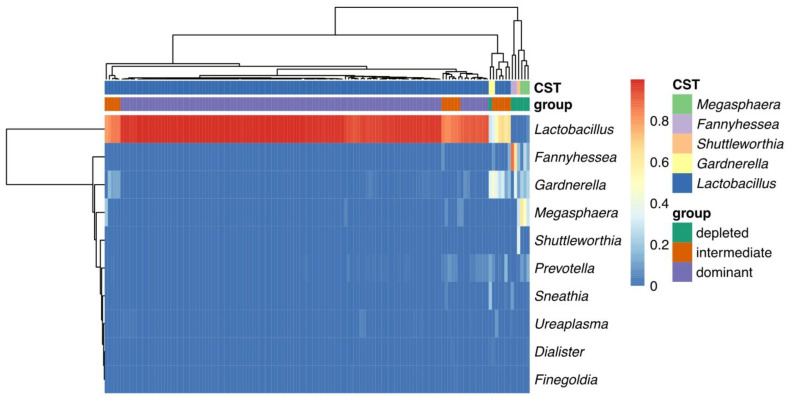
Integrated analysis of vaginal microbiome in all the samples. The heat map clustering of vaginal bacteria composition in the three groups.

**Table 1 pathogens-11-00847-t001:** Clinical characteristics of the subjects.

Characteristics	Whole Study Cohort (*n* = 441)			16S rDNA Amplicon Sequencing (*n* = 135)		
	PROM Cases (*n* = 84, 19.05%)	Controls (*n* = 342, 77.55%)	Lost to Follow-Up (*n* = 15, 3.40%)	PROM Cases (*n* = 45)	Controls (*n* = 90)	*p*-Value
Maternal age ^a^	29.06 ± 4.08	29.91 ± 4.41	29.67 ± 4.3	28.96 ± 3.80	29.70 ± 3.66	0.251 ^†^
Nulliparity	55 (65.48%)	180 (52.63%)	10 (66.67%)	32 (71.11%)	49 (54.44%)	0.09
White blood cell at admission (×10^9^/L)	9.82 ± 2.8	9.4 ± 2.07	9.4 ± 1.65	10.01 ± 3.00	9.49 ± 2.67	0.242 ^†^
Positive genital cultures at admission	24(28.57%)	57(16.67%)	4 (26.67%)	7 (15.56%)	12 (13.33%)	0.795
Gestational age at sampling (weeks) ^a^	32.84 ± 1.53	32.54 ± 1.43	32.81 ± 1.47	33.18 ± 1.7	33.01 ± 1.59	0.553 ^†^
Steroid administration	6 (7.14%)	5 (1.46%)	-	0	0	-
Tocolysis treatment	17 (20.24%)	61 (17.83%)	-	8 (17.78%)	7 (7.78%)	0.099 ^‡^
Gestational age at delivery (weeks) ^a^	38.46 ± 1.60	38.93 ± 1.05	-	39.27 ± 1.45	39.65 ± 0.97	0.098 ^†^
Latency from sampling to delivery (weeks) ^a^	5.16 ± 6.23	6.08 ± 2.00	-	5.84 ± 2.36	6.39 ± 0.14	0.148 ^†^
BMI in the first trimester(kg/m^2^) ^a^	22.15 ± 2.90	22.23 ± 4.98	-	22.19 ± 3.31	21.74 ± 2.73	0.140 ^†^
BMI in third trimester (kg/m^2^) ^a^	26.36 ± 6.33	26.89 ± 5.16	-	27.30 ± 3.84	27.66 ± 2.98	0.537 ^†^
Estriol at sampling ^a^	7.22 ± 0.44	7.61 ± 1.66	-	7.22 ± 2.79	7.38 ± 2.78	0.748 ^†^
Number of deliveries ^a^	1.29 ± 0.17	1.44 ± 0.54	-	1.31 ± 0.51	1.47 ± 0.56	0.101 ^†^
spontaneous abortion ^a^	0.91 ± 0.05	0.67 ± 0.73	-	1.00 ± 0.16	0.66 ± 0.81	0.053 ^†^
Baby weight at birth (g) ^a^	3318.78 ± 525.96	3397.95 ± 441.45	-	3031 ± 368.75	3041 ± 379.09	0.168 ^†^
Baby gender (male/female)	48/36 (57.14%/42.86%)	226/106 (66.08%/30.99%)	-	25/20 (55.56%/44.44%)	48/42 (53.33%/46.67%)	0.84 ^‡^
Spontaneous abortion (number) ^a^	0.93 ± 1.12	0.67 ± 0.73		1.00 ± 1.15	0.66 ± 0.91	0.053 ^†^
gestational diabetes	15 (17.86%)	41 (11.99%)	-	0	0	-
preeclampsia	16 (19.05%)	38 (11.11%)		0	0	-

PROM: premature rupture of membranes. ^a^ Values are presented as median ± SEM. ^†^ Independent-sample Mann-Whitney U test. ^‡^ Fisher’s Exact Test.

**Table 2 pathogens-11-00847-t002:** Integrated analysis of vaginal microbiome in all the samples. Logistic regression model analysis identified vaginal bacteria that could predict the risk of PROM development.

	Estimated Coeffieicent (95% CI)	Adjusted *p*-Value
*Lactobacillus*	−0.09 (−0.06 to −0.01)	0.04
*Fannyhessea*	−0.01 (−0.05 to 0.03)	0.57
*Gardnerella*	0.04 (0.01 to 0.06)	0.02
*Megasphaera*	0.04 (0.01 to 0.06)	0.01
*Shuttleworthia*	−0.004 (−0.02 to 0.007)	0.48
*Prevotella*	0.11 (0.002 to 0.02)	0.02
*Sneathia*	0.005 (−0.002 to 0.01)	0.16
*Ureaplasma*	0.004 (0.001 to 0.007)	0.01
*Dialister*	0.001 (0.001 to 0.0003)	0.04
*Finegoldia*	−0.47 × 10^−5^ (−0.0009 to 0.0008)	0.93

## Data Availability

The datasets utilized and/or analysed during the current study have been available from the corresponding author on reasonable request. Raw data of 16S rDNA sequencing was deposited to Sequence Read Archive (SRA number: PRJNA860958. https://www.ncbi.nlm.nih.gov/bioproject/PRJNA860958, accessed on 11 July 2022).

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
