# Peer review of "Characterization of Vaginal Microbiota in Third Trimester Premature Rupture of Membranes Patients through 16S rDNA Sequencing"

_pathogens, 2022, doi:10.3390/pathogens11080847_

Round 1
Reviewer 1 Report
The manuscript by Lou Liu et al. describes interesting data on characterizing the vaginal microbiota during the 3rd trimester of PROM using 16S rDNA sequencing.
The manuscript has clearly been improved by the review in accordance with my previous comments, as I did not recommend rejection but a major revision in my second comments. I recommend that authors do not consider my previous comments unfair.
Two of my comments were not addressed appropriately.
First, the Harvard database cannot be considered sufficient. Please consider depositing your data on SRA, giving an appropriate SRA number, as most sequencing papers have done. Also, note that the URL link may not work depending on the nature of the PCR or software, limiting the upload of data for a meta-analysis for example.
I strongly disagree with BGI, as it is necessary to have a control (negative, as performed) but also a positive control, to validate detection of all appropriate strains. Many published studies suggest/demonstrate this important need. Please consider
Author Response
The manuscript by Lou Liu et al. describes interesting data on characterizing the vaginal microbiota during the 3rd trimester of PROM using 16S rDNA sequencing.
The manuscript has clearly been improved by the review in accordance with my previous comments, as I did not recommend rejection but a major revision in my second comments. I recommend that authors do not consider my previous comments unfair.
We apologize for our early improper language. We highly appreciate the opportunity to further improve our manuscript. Thank you!
Two of my comments were not addressed appropriately.
First, the Harvard database cannot be considered sufficient. Please consider depositing your data on SRA, giving an appropriate SRA number, as most sequencing papers have done. Also, note that the URL link may not work depending on the nature of the PCR or software, limiting the upload of data for a meta-analysis for example.
We thank the Reviewer for the good suggestion. We have now deposited our data to SRA and provided access number in the revised manuscript.
Lines: 564-566. “Raw data of 16S rDNA sequencing was deposited to Sequence Read Archive (SRA number: PRJNA860958. https://www.ncbi.nlm.nih.gov/bioproject/PRJNA860958).”
I strongly disagree with BGI, as it is necessary to have a control (negative, as performed) but also a positive control, to validate detection of all appropriate strains. Many published studies suggest/demonstrate this important need. Please consider
We thank the Reviewer for raising this technical question. We strongly agree with the Reviewer that both negative and positive controls are critical in genomic sequencing studies. We had proper negative control in our study however a positive control showing the capability of detecting all related bacterial strains was missing. We apologize for this mistake and will include this kind of control in our future studies. However, according to our current result, all major vaginal infection related strains were detected indicating the robustness of the sequencing methodology.
To further understand the importance of positive control in genomic sequencing studies, we ran an extensive literature search and did find studies with such controls [1], at the same time, many studies did not have a proper positive control, including those done by BGI [2] [3,4]as well as other institutions [5-7]. We highly appreciate the Reviewer’s suggestion and will definitely have such control in our future studies. But please forgive us this time, as this will not impair our final conclusion in this manuscript.
- Kitaya, K.; Nagai, Y.; Arai, W.; Sakuraba, Y.; Ishikawa, T. Characterization of Microbiota in Endometrial Fluid and Vaginal Secretions in Infertile Women with Repeated Implantation Failure. Mediators Inflamm 2019, 2019, 4893437, doi:10.1155/2019/4893437.
- Chen, C.; Song, X.; Wei, W.; Zhong, H.; Dai, J.; Lan, Z.; Li, F.; Yu, X.; Feng, Q.; Wang, Z.; et al. The microbiota continuum along the female reproductive tract and its relation to uterine-related diseases. Nat Commun 2017, 8, 875-885, doi:10.1038/s41467-017-00901-0.
- ZongchaoMo; PeideHuang; ChaoYang. Meta-analysis of 16SrRNA Microbia lData Identifified Distinctive and Predictive Microbiota Dysbiosis in Colorecta lCarcinoma Adjacent Tissue. mSystems 2020, 5, 1-16, doi:10.1128/mSystems.
- Liu, Z.; Dai, X.; Zhang, H.; Shi, R.; Hui, Y.; Jin, X.; Zhang, W.; Wang, L.; Wang, Q.; Wang, D.; et al. Gut microbiota mediates intermittent-fasting alleviation of diabetes-induced cognitive impairment. Nat Commun 2020, 11, 855, doi:10.1038/s41467-020-14676-4.
- Feehily, C.; Crosby, D.; Walsh, C.J.; Lawton, E.M.; Higgins, S.; McAuliffe, F.M.; Cotter, P.D. Shotgun sequencing of the vaginal microbiome reveals both a species and functional potential signature of preterm birth. NPJ Biofilms Microbiomes 2020, 6, 1-9, doi:10.1038/s41522-020-00162-8.
- Fettweis, J.M.; Serrano, M.G.; Brooks, J.P.; Edwards, D.J.; Girerd, P.H.; Parikh, H.I.; Huang, B.; Arodz, T.J.; Edupuganti, L.; Glascock, A.L.; et al. The vaginal microbiome and preterm birth. Nat Med 2019, 25, 1012-1021, doi:10.1038/s41591-019-0450-2.
- Brown, R.G.; Marchesi, J.R.; Lee, Y.S.; Smith, A.; Lehne, B.; Kindinger, L.M.; Terzidou, V.; Holmes, E.; Nicholson, J.K.; Bennett, P.R.; et al. Vaginal dysbiosis increases risk of preterm fetal membrane rupture, neonatal sepsis and is exacerbated by erythromycin. BMC Med 2018, 16, 1-15, doi:10.1186/s12916-017-0999-x.

Reviewer 2 Report
Reviewer Report
Comments and Suggestions for Authors - pathogens-1839034 (previously ID 1772563 before major revisions)
Congratulations to the authors for the excellent work. Major revisions were done in the original manuscript. The manuscript is now more clear and well-written.
The authors rectified the manuscript taking in consideration all my previous comments, although I did not understand why the manuscript changed its ID code (previously ID was 1772563 and now its ID is1839034) and no reply to Reviewer’s comments was received.
I will endorse the revised version of the manuscript after some minor revisions, which I put in my comments below.
Minor comments
Abstract
Line 33-34- Please clarify the reason of the repetition of coeff and p-values from the genera previously described on lines 31-32, otherwise remove this repetition.
Introduction
Line 52- Please rectify the “Aggressive vaginal infection by Peptostreptococcus, Streptococcus and Ureaplasma etc.” with “Aggressive vaginal infection by Peptostreptococcus, Streptococcus and Ureaplasma among others”.
Line 58- Please rectify “Strep-tococcus”.
Line 71-72- Please clarify the sentence, I did not understand the meaning of the sentence or “term PROM”. Do you mean preterm PROM (pPROM) right?
Line 74- Please add range of preterm PROM rates reported worldwide and cite the references in “… what has been reported worldwide …”.
Line 76- Please rectify “… certain strains of staphylococcus, such as some strains of Staphylococcus aureus…”.
Line 90- Please rectify “characterise" with “characterize”.
Results
Lines 97-107- Please check the different letter size on the text of the subsection “2.1. Demographical and clinical data”.
Lines 291-292- Please put the title of the Table 2 before the respective table on line 289.
Discussion
Line 330- Please rectify “Lacto-bacillus” with “Lactobacillus spp.”
Lines 336-337- Please replace “Pacha-Herrera [24] et al identified…” with “Pacha-Herrera et al [24] identified…”.
Line 337- Please replace “… and found L. acidophilus …” with “… and found that L. acidophilus …”.
Line 367- Please rectify “unreaplasma” with “Ureaplasma spp.”
Line 381- Please rectify “Lacto-bacillus” with “Lactobacillus spp.”
Lines 408-410- Please rectify “… approval for the study, which was carried out in accordance with the Declaration of Helsinki Principles (No. 2019-02-0402-01) and conducted following the Declaration of Helsinki Principles.” with “… approval for the study (No. 2019-02-0402-01) and the present study was conducted following the Declaration of Helsinki Principles.”.
Conclusions
Lines 502-503- Please clarify what “high expression” was connected to the vaginal microbiota in PROM cases.
Lines 505-507- Please rectify the sentence, for example, by replacing “… , which could result into PROM.” with “… , a PROM case is easily established among women”.
Lines 507-510- Please simplified the two sentences without repeating the same 3 genera. For example: Replace “The mainly pathogens in the PROM group were Gardnerella, Prevotella, and Megasphaera at the genus level in this study. Those pathogens which included Prevotella, Gardnerella and Megasphaera were associated with the increased risk of PROM.” with “The mainly pathogens in the PROM group were Gardnerella, Prevotella, and Megasphaera at the genus level in this study, being statistically associated with PROM establishment among women.”.
Lines 510-511- Please do not write “viruses”. Bacteria are not virus and therefore the term viruses cannot be applied in conclusions. Please rectify the sentence, for example: “Further research is needed to understand how these opportunistic pathogens interact with the host and the vaginal microbiome”.
Also, state what kind of research should be realized in the future. The sentence is too vague.

Author Response
Comments and Suggestions for Authors
Reviewer Report
Comments and Suggestions for Authors - pathogens-1839034 (previously ID 1772563 before major revisions)
Congratulations to the authors for the excellent work. Major revisions were done in the original manuscript. The manuscript is now more clear and well-written.
The authors rectified the manuscript taking in consideration all my previous comments, although I did not understand why the manuscript changed its ID code (previously ID was 1772563 and now its ID is1839034) and no reply to Reviewer’s comments was received.
I will endorse the revised version of the manuscript after some minor revisions, which I put in my comments below.
Thank you again for your constructive comments helping us improve the manuscript. We appreciate your hard work to locate all those typos and technical mistakes, as this is really helpful to us as non-English speaking authors. We have gone through the manuscript carefully and corrected all these mistakes.
Minor comments
Abstract
Line 33-34 Please clarify the reason of the repetition of coeff and p-values from the genera previously described on lines 31-32, otherwise remove this repetition.
Apologize for the confusion. We believe this was due to multiple editing tracks. We therefore removed those repetitive sentences. Lines 34-36.
Introduction
Line 52- Please rectify the “Aggressive vaginal infection by Peptostreptococcus, Streptococcus and Ureaplasma etc.” with “Aggressive vaginal infection by Peptostreptococcus, Streptococcus and Ureaplasma among others”.
Fixed as suggested. Lines 54-56.
Line 58- Please rectify “Strep-tococcus”.
Might be software glitch. Fixed. Line 61.
Line 71-72- Please clarify the sentence, I did not understand the meaning of the sentence or “term PROM”. Do you mean preterm PROM (pPROM) right?
“Xia et al. [8] carried out another large population study and discovered 17049 (15.3%) women with PROM and among which 13927 (12.5%; and 81.7% of all PROM cases) were term PROM.”
Sorry for confusion. We meant these PROM cases happened when close to term (31-36+6 weeks, in comparison to pre-term, <37 weeks). The word “term PROM” was from original paper cited. The language may be not accurate but we did not want to change it. It is due to the fact that “term PROM” is more common than pre-term PROM in China, which we also explained the reason in the original manuscript.
Lines 71-74. “Due to China’s slightly different definition of preterm birth (delivery between 24 and 36+6 gestational weeks), plus the popularity of tocolytics in clinical practice, the preterm birth rate had been reported as low as 6.9% in a recent epidemiology study in 2015-2016 [9].”
Line 74- Please add range of preterm PROM rates reported worldwide and cite the references in “… what has been reported worldwide …”.
In China, preterm PROM rate is about 2.6% [10], while it has been reported as 3-3.25% in United States [11]. So we added this number and reference in the revised manuscript, and changed “worldwide” to “United States” to make it more accurate.
Lines 76-80. “According to these studies, although PROM rate is high, the preterm PROM rate is comparatively low in Chinese population (2.6%) [10] in contrast to what has been reported in United States (3-3.25%) [11] , yet, the unfavourable pregnancy outcomes are comparable.”
Line 76- Please rectify “… certain strains of staphylococcus, such as some strains of Staphylococcus aureus…”.
Fixed. Line82.
Line 90- Please rectify “characterise" with “characterize”.
Fixed. Line 95.
Results
Lines 97-107- Please check the different letter size on the text of the subsection “2.1. Demographical and clinical data”.
Fixed. Lines 103-114.
Lines 291-292- Please put the title of the Table 2 before the respective table on line 289.
Fixed. Lines 296-297.
Discussion
Line 330- Please rectify “Lacto-bacillus” with “Lactobacillus spp.”
Fixed. Line 341.
Lines 336-337- Please replace “Pacha-Herrera [24] et al identified…” with “Pacha-Herrera et al [24] identified…”.
Fixed. Line 348.
Line 337- Please replace “… and found L. acidophilus …” with “… and found that L. acidophilus …”.
Fixed. Line 349.
Line 367- Please rectify “unreaplasma” with “Ureaplasma spp.”
Fixed. Line 378-379.
Line 381- Please rectify “Lacto-bacillus” with “Lactobacillus spp.”
Fixed. Line393-394.
Lines 408-410- Please rectify “… approval for the study, which was carried out in accordance with the Declaration of Helsinki Principles (No. 2019-02-0402-01) and conducted following the Declaration of Helsinki Principles.” with “… approval for the study (No. 2019-02-0402-01) and the present study was conducted following the Declaration of Helsinki Principles.”.
Fixed. Lines 423-424.
Conclusions
Lines 502-503- Please clarify what “high expression” was connected to the vaginal microbiota in PROM cases.
Fixed. Line 518-520. “Low levels of Lactobacillus and high levels and diversity of conditional pathogenic bacteria were connected with the characteristics of the vaginal microbiota in PROM cases.”
Lines 505-507- Please rectify the sentence, for example, by replacing “… , which could result into PROM.” with “… , a PROM case is easily established among women”.
Fixed. Lines 524.
Lines 507-510- Please simplified the two sentences without repeating the same 3 genera. For example: Replace “The mainly pathogens in the PROM group were Gardnerella, Prevotella, and Megasphaera at the genus level in this study. Those pathogens which included Prevotella, Gardnerella and Megasphaera were associated with the increased risk of PROM.” with “The mainly pathogens in the PROM group were Gardnerella, Prevotella, and Megasphaera at the genus level in this study, being statistically associated with PROM establishment among women.”.
Fixed. Lines 527-529.
Lines 510-511- Please do not write “viruses”. Bacteria are not virus and therefore the term viruses cannot be applied in conclusions. Please rectify the sentence, for example: “Further research is needed to understand how these opportunistic pathogens interact with the host and the vaginal microbiome”.
Fixed. Line531.
Also, state what kind of research should be realized in the future. The sentence is too vague.
Thanks for the suggestion. We have added one example in the revised manuscript. Lines529-532. “Further research, such as co-culture experiment using fetal membrane tissue and bacteria identified in this study, is needed to better understand how these conditional pathogens interact with the host and the vaginal microbiome and result in PROM.”

Round 2
Reviewer 1 Report
The authors have successfully improved the manuscript according to my preivous comments and is now suitable for publication.
This manuscript is a resubmission of an earlier submission. The following is a list of the peer review reports and author responses from that submission.
Round 1
Reviewer 1 Report
This manuscript by Lou Liu et al. characterizes the vaginal microbiota in the third trimester to identify biomarkers for PROM outcome.
This manuscript is of low interest and presentation because it is supported by unsuitable data (V3-V4) when studies have shown that V1-V3 are more appropriate. In addition, the data are not very well presented and deserve to be strongly reviewed and supplemented.
Global: italicize "et al.", microbial names, versus.
Results: A flow chart is needed to understand the selection process.
Table 1: percentage present by category for each parameter.
Typo: refraction: rarefaction.
Figure 1 is pixelated.
Figure 2 is pixelated but inverted
Figure 4b is a table and not a figure.
Methods: justify the number of patients included.
Part 4.4: justify the selection of patients.
Method: is a positive control added to the trial?
Method: is the sequencing done in 300*2 paired end?
What was the number of multiplexing?
The version of the Silva database should be referenced appropriately.
Author Response
We thank the Editor and Reviewer for giving us the opportunity to revise our manuscript. These comments greatly helped us improve the quality of the manuscript. We’ve made substantial revision to the original manuscript and addressed all Reviewers comments, and here we also provided point-to-point response to each comment.
Please see the attachment

Reviewer 2 Report
Congratulations to the authors for the excellent work.
The manuscript is clear and well-written, showing a research design to validate the main goal of the study.
Please take in consideration that in the vaginal microbiota, now Fannyhessea vaginae is the current species for the species previously known as Atopobium vaginae, as you may consult in:
https://www.ncbi.nlm.nih.gov/Taxonomy/Browser/wwwtax.cgi?mode=Info&id=82135
Therefore, the authors should mention this information in the manuscript to avoid future critics by peers.
Also, it is not only “different Lactobacillus species may play very different role in vaginal microenvironment” but also the consortia formed by different lactobacilli. Please add this information in the Discussion section and you can cite several papers (such as: https://doi.org/10.3389/fcimb.2022.863208).
Finally, I recommended the publication of the manuscript after minor revisions.
Minor comments
Abstract
Lines 25-26: Please rectify the term “Lactobacillus abundance--dominant”.
Line 31 and 34: Please rectify the term “pathogenic bacterial” with “pathogenic bacteria”.
Line 32: Please delete the extra space after “Ureaplasma”.
Line 34: Please specify all BV-related genera as such, by replacing “bacterial” with genera.
Introduction
Line 45: Please add “a” before “mixed infection”.
Lines 51 and 52: Please indicate some examples of “Aggressive vaginal infection releases bacterial toxins and inflammatory cytokines”.
Lines 55-57: Please rectify the confusing statement “When microbiome balance in the vaginal is disrupted, mixed vaginal infection can cause microbiome dysbiosis, leading to adverse pregnancy outcomes, such as group B Streptococcus infection[6] and vulvovaginal candidiasis[7].” A vaginal infection or mixed vaginal infection is already a dysbiosis state, however a vaginal dysbiosis is not always a vaginal infection or commonly known vaginitis. Please be careful with these statements that prejudice your manuscript.
Lines 57-59: Candida albicans and Trichomonas should be in italics. Also, the authors meant Trichomonas vaginalis, right? Gardnerella should be followed by “species”.
Line 59: “Lactobacilli” is not “Lactobacillus” and therefore it should not be in italics or even written with capital L. Once again, these errors are harming your manuscript. Please realize a serious English editing of the manuscript.
Line 71: Delete the extra space in the sentence.
Lines 74-75: Please specify some examples of “certain antibiotics resistant bacterial strains [13]”.
Results
Line 121: Please add the comma before “respectively”.
Lines 124-125: Please replace “(P=0.014, P=0.014, respectively FigS2c and S2d)” with “(both P=0.014 values; see Fig S2c and Fig S2d)”. Also, I recommend the authors to realize a further English editing in the remaining manuscript, since I can not appoint all syntax errors in the text.
Line 138: Please put also “Dialister” in italics.
Line 164: Please replace “microenvironment” with “microbiota” or “microbiome”, also replace the term in the remaining manuscript.
Line 176: Please take in consideration that in the vaginal microbiota, now Fannyhessea vaginae is the current species for the species previously known as Atopobium vaginae, as you may consult in:
https://www.ncbi.nlm.nih.gov/Taxonomy/Browser/wwwtax.cgi?mode=Info&id=82135
Therefore, the authors should mention this information in the manuscript to avoid future critics by peers.
Line 198: Please replace “flora” with “microbiota” or “microbiome”, also replace the term in the remaining manuscript (such as line 260 in Discussion section).
Line 203: Please replace “principal co-ordinates analysis (PCoA)” with “Principal Coordinates Analysis (PCoA)”, also replace the term in the remaining manuscript.
Line 207: Please replace “(P=0.02 Fig S3e)” with “(P=0.02; see Fig S3e)”. Also, the authors should rectify similar references in the remaining manuscript.
Line 210: Please replace “(P=0.07, P=0.04 respectively, Fig S4c, Fig 210 S4d)” with “(P=0.07 and P=0.04, respectively; see Fig S4c and Fig S4d)”. Also, the authors should rectify similar references in the remaining manuscript.
Line 216: Please put “versus” in italics, also confirm the italics form in the remaining manuscript.
Line 246: Please put the genera names in italics, also check the remaining manuscript. The same non-italics forms of the genera names are present in the Figure 4.
Discussion
Lines 294-296: Please add references to the following sentence: “It was worth noting that 16S rDNA sequencing technology is not able to clearly differentiate different Lactobacillus species, while those different Lactobacillus species may play very different role in vaginal microenvironment.”
I suggest the following reference that analyze the probiotic potential of different Lactobacillus species against BV and Vaginal dysbiosis: https://pubmed.ncbi.nlm.nih.gov/35646732/ or https://doi.org/10.3389/fcimb.2022.863208
Pacha-Herrera D, Erazo-Garcia MP, Cueva DF, Orellana M, Borja-Serrano P, Arboleda C, Tejera E, Machado A. Clustering Analysis of the Multi-Microbial Consortium by Lactobacillus Species Against Vaginal Dysbiosis Among Ecuadorian Women. Front Cell Infect Microbiol. 2022 May 11;12:863208. doi: 10.3389/fcimb.2022.863208.
Also, it is not only “different Lactobacillus species may play very different role in vaginal microenvironment” but also the consortia formed by different lactobacilli. Please add this information in the Discussion section. Finally, replace “microenvironment” with “microbiota”, as previously commented.
Lines 299-300: Please put “L.” in italics. Check the remaining text (such as line 306).
Line 318: Please remove the space before the comma after “Ureaplasma”.
Methods
Lines 405-406: Please add the references of the software programs used in this study “The software used for sequence processing and OTU clustering includes: Trimmomatic, FLASH, QIIME version 2.0, Dereplication, and Singletons”.
Line 415: Please add reference in the sentence “Maternal characteristics were analyzed in the R environment”.
Line 424: Please add the space before “[36]”, also check for this type of mistake in the remaining manuscript (I also found the same mistake before).
Author Response
We thank the Editor and Reviewer for giving us the opportunity to revise our manuscript. These comments greatly helped us improve the quality of the manuscript. We’ve made substantial revision to the original manuscript and addressed all Reviewers comments, and here we also provided point-to-point response to each comment.
Please see the attachment.

Reviewer 3 Report
As a general comment on the structure of the paper, it would benefit from the following organization: Introduction, Material and Methods, Discussion and Conclusion rather than the structure presented.
Introduction
The introduction is consistent and well supported by the literature. The authors provide a detailed description of the PROM concept and the microbiome environment at the time of membrane rupture. However, the authors often refer to the concept of mixed infection (between lines 50 and 56) as a justification for PROM. However, this paragraph needs to be reorganized so that the information is more consistent and not so repetitive.
Line 59 -61 "Vaginal Lactobacilli maintain the balance of the local microbial environment by decreasing vaginal pH through the production of lactic acid and reduce the risk of pregnancy complications." This statement will make more sense before referencing the imbalances that can occur in the vaginal microenvironment.
The authors should mention the Lactobacillus species associated with the healthy vaginal microenvironment as it becomes relevant to the paper's discussion.
The authors have defined the purpose of the study clearly.
Materials and methods
Lines 346 - 355 - Regarding the study population, the authors should indicate the age range of the women included in the study, as the risk of PROM may be associated with their age. Alternatively, refer to table 1.
In the materials and methods, the authors have repeated topics 4.2 and 4.3 (lines 356 - 373). The authors should remove topic 4.3 and standardize the numbering.
It is not clear in the extraction protocol (4.5. DNA extraction) if the structure of Gram-positive bacteria was considered. If the authors have used lysozyme as part of the extraction process, they should indicate that this process step is critical for library preparation and successful amplification of the microorganisms.
Line 389 - 394. The authors do not mention the enzyme or master mix used to amplify the 16S rRNA products. Instead, the authors should indicate the reagents used for the library preparation.
Results
The authors state, "We had difficulty to determine about 50% of the species when using QIIME 2.0 software to analyze flora composition at the ASV-level" (lines 130-131). Did the authors use a sufficiently trained and curated database for species-level identification of the microbiome?
Figure 1e. The authors should correct the image resize and make the panel of images that make up Figure 1 uniform.
Figure 2 is inverted. The authors should place it correctly. In addition, panel d of figure 2 (cladogram) should be adjusted, so it is possible to read its content correctly.
Authors should place the species names in italic throughout the paper, with particular attention to Figure 4.
Discussion and conclusions
The authors conduct a discussion well supported by the new results obtained in this study and the literature. However, one would expect that the hypervariable region would allow a species-level characterization for some of the genera mentioned. It will be essential for the authors to mention possible limitations that generated this limitation in the 16S rRNA metagenome analysis (experimental or bioinformatics analysis limitations).
Author Response

(The authors gave the same response as above.)

Round 2
Reviewer 1 Report
The authors have revised the manuscript on the basis of my previous comments, as far as they can.
Numerous typos remains.
Moreover, please indicate the SRA number for the analysed data (and not "on reasonable request').
Author Response
The authors have revised the manuscript on the basis of my previous comments, as far as they can.
we thank the reviewer for these comments.
Numerous typos remains.
We went through the manuscript again carefully to correct all typos.
Moreover, please indicate the SRA number for the analysed data (and not "on reasonable request').
We have included the link of our raw data to Harvard dataverse at the end of our revised manuscript.
https://dataverse.harvard.edu/dataset.xhtml?persistentId=doi:10.7910/DVN/W3J9R3&version=1.0